# Economic Development and Changes in Human Resource Management in a Sustainable Agricultural Sector: Recent Evidence from Brazilian Sugar–Alcohol Companies

**Paulo Mourao** [1,*], **Edson Kubo** [2], **Isabel Santos** [2] and **Valeria Mazucato** [2]

1   Department of Economics & NIPE, University of Minho, 4700-000 Braga, Portugal
2   Graduate Course of Management Departament, University of Sao Caetano do Sul (USCS), Campus Conceicao, Sao Caetano do Sul 09530-060, Brazil; edson.kubo@prof.uscs.edu.br (E.K.); isabel.santos@prof.uscs.edu.br (I.S.); mazasun@gmail.com (V.M.)
*   Correspondence: paulom@eeg.uminho.pt

**Abstract:** Economic development causes significant changes in the innovative processes of resource management. The purpose of this article is to discuss the sustainability of resource management in sugar–alcohol companies operating in the most productive areas of Brazil and to analyze the profile of the companies in the sector in terms of innovation and adherence to practices of human resource management. The methodological procedures were based on qualitative research. Data collection was directed toward the population of companies in the sector of cane brandy production, having as the main criteria representativeness of human resource management and to be active. The results indicate that human management tends to be traditional in spite of several current economic challenges. The conclusions also highlight that the most highly mechanized companies are those that have adhered to strategic policies and practices, and that the traditional model of human management prevails in more than 70% of companies.

**Keywords:** sustainability; economic development; sugar and alcohol sector; cane brandy

## 1. Introduction

For a country with agricultural traditions and whose economic axis is based on the production of commodities, Brazil has been experiencing technological evolution in the production and management of agribusiness. Productivity has been increasing with the adoption of soil management and remediation techniques, the use of georeferencing systems, and embedded electronics [1]. Change is, in fact, paradigmatic but not recent. The green revolution—an effort to increase global food production—that reached Mexico, India, and Pakistan in the 1950s began in Brazil in the 1980s and continued more effectively over the past 20 years; according to Pereira et al. [2], this can be translated to rural producers as a necessity of structuring agricultural production with an emphasis on the decision-making process and efficient management of natural, human, physical, and financial resources.

The professionalization of agribusiness expands towards the qualification of the workforce of the rural sector, leaving in the past the stereotype of disqualified and contingent labor. The rural worker has become an operator of sophisticated equipment and systems, an analyst of information and of prospecting models. Therefore, technological advances have repercussions on the model of human resource management (HRM) throughout the entire agribusiness production chain that covers

all stages of agricultural production, from the field to commercialization, including the intermediate processes of processing, packaging, transportation, and distribution, as well as the knowledge of certification standards [3].

This article emerges from research that aimed to examine how HRM is characterized in the sugar and alcohol companies in the state of Mato Grosso do Sul (which are predominantly responsible for the famous production of Brazilian cane brandy). According to the National Supply Company [4], the sector of sugar and alcohol companies accounts for 7.09% of the volume produced in the country. The state of Mato Grosso do Sul is the fourth-largest producer of sugarcane in the country, fifth for sugar, and fourth for ethanol in the country.

The general objective of this research is therefore to evaluate how the processes of economic development have caused changes in the HRM performed in sugar and alcohol companies in Brazil. As specific objectives, which will allow for the achievement of the general objective of the research, the following stand out:

- To discriminate the main impacted fields of action of HRM and, therefore, to establish items to verify their characteristics;
- To check in the field (i.e., in the sugar and alcohol companies of Brazil), how the stimuli from the processes of economic development have led to significant modifications (or not) in the characteristics studied by HRM.

The article has four sections. The second section presents an overview of the research and the context of its realization, which aims to describe HRM configurations in the sugar–alcohol companies operating in the state of Mato Grosso do Sul. It also describes the main theoretical elements that support the operational concepts and research assumptions. The third section describes the adopted research procedures and discusses the results in relation to the theoretical framework. In the final section, the conclusions are presented along with a discussion of the research assumptions.

## 2. Literature Review about Economic Development, Innovation, and Resistant Agents

The processes of economic development are understood in this work as processes that, in line with Perroux [5], introduce changes in the way the production, the distribution of resources, and the results of efforts, as well as the consumption of communities, are processed. These changes are social processes that involve a deep dynamic of the communities.

Like Perroux [5] stressed, communities are composed of networks of heterogeneous individuals who, with individual differences, still increase the diversity of responses with the pace of economic development. There are communities that, due to their own characteristics—such as the predominance of younger groups—are more receptive to the processes of globalization, with less investment made and with the respective rates of return influenced by the change in development, becoming more favorable supporters of new methods and more easily adhering to change.

However, other communities are traditionally more resistant to change. As Haberler [6] showed, there are communities in which the predominant groups, but also the leaders, reveal a less clear perception of the gains to be obtained with the change. The reasons for this resistance can range from the median ages in the community and of leaders, to the difficulty of perceiving advantages for change and for economic development.

From the dynamics of the diversity of these groups and communities, it can be seen how the processes of change associated with economic development share three conditions: they are long, involve tensions (that is, difficulties of individual and group responses), and are hybrid, in the concept of Shepard [7] (that is, there will always be individuals in any community that are different from the rest of the group, and there is not one individual who in all dynamics of innovation is always resistant or a promoter of all the changes).

In terms of economic sectors and their companies, the reactions are also divided between resistance to innovation or adhesion [7]. For example, Walker [8] showed that there is a certain

tendency towards resistance to innovation from sectors and agents more closely linked to the primary sector or agricultural activities. Explanations are explored in the works of Gouldner [9] and Walker [8] and include the traditional profile of the owner and entrepreneur of this sector and aspects of organizational culture that affect companies and exporting companies in the sector, even in recent periods [10].

The issue of sustainability in development processes cannot be separated from the profiles of economic actors. At various times, economic sustainability collides with innovation (and vice versa); but also at other times, development, sustainability, and innovation are moving in the same direction. Therefore, it is essential to increase sustainability both in the economy and in the social mechanisms themselves, to understand the way in which economic agents position themselves in the face of innovation and change processes.

As Brazil is a country with a significant capacity for the production and export of goods from the primary sector, this work aims to analyze the predominance of the agents in the sector. We intend to analyze if the sector is dominated by promoters of change or by those who are resistant to change. We are going to particularly focus in the management of human resources, a dimension particularly sensitive to the level and significance of change. The next sub-section describes the main theoretical contributions that support the construction of our research assumptions. Given the relevance to the national GDP, we begin by outlining the main features of the evolution of the sugar and alcohol industry, later reflecting on human resource management in this industry, raising the central hypothesis that the sector, despite modernization, still reveals the characteristics of traditional human resource management.

The next sections will carry out this analysis.

## 2.1. The Brazilian Sugarcane Industry—Importance and Evolution

Brazil's gross domestic product (GDP) for the period between 2017 and 2019 has been positively influenced by agribusiness activities. There was a growth of 6.1% in agriculture and livestock, 2.7% in industrial activity, and 1.7% in the services sector [11,12].

The economic strength of Brazilian agribusiness is reflected in the results of 2017, when the agribusiness sector accounted for 41% of the volume exported that year and 22% of the wealth generated in the country, reaching R $1.42 trillion (about EUR 300 billion), according to data from the Brazilian Association of Meat-Exporting Industries [13]. Of this value, approximately 69% corresponds to the productive chain of the agricultural branch. By 2018, the agribusiness production chain presented even more robust results, with a growth estimate of 2.36% [14].

Although the production of agribusiness is dispersed throughout Brazil, five states are often considered the main export producers in the sector. In 2015, the states of São Paulo, Mato Grosso, Rio Grande do Sul, Paraná, and Minas Gerais accounted for 67.77% of the volume of agricultural products exported by the country. In 2018, the ranking of the states positioned Mato Grosso, São Paulo, Paraná, Minas Gerais, and Rio Grande do Sul as responsible for the production of 59.7% of the total produced by this sector. Soybeans, sugarcane, corn, herbaceous cotton, and coffee generated 80% of the value of the crops. In this group, soybeans stood out with the highest value generated—R $143.5 billion in 2018, around EUR 33 billion [14].

Food production has been considered one of the strategic areas for national development and is one of the fundamental components of the national strategy for science, technology, and innovation, together with energy, defense, health, and others for the years that extend to 2022. The contributions of new technologies have reduced the occupancy rates of the workforce and allocated more qualified human resources for field activities. It is, therefore, relevant to question the practices and policies of personnel management in a sector previously characterized by the existence of less-qualified labor.

In turn, the sugar and alcohol industries have experienced remarkable growth both in revenues and in the application of new technologies. The sector is responsible for more than R $40 billion per year (EUR 10 billion) and generates about 3.6 million jobs [15]. In Mato Grosso do Sul, the regional locus of this research, sugarcane cultivation increased the planting area from 6 to 39 producing cities

over the last decade, predominating in the economy of 50% of the municipalities of Mato Grosso do Sul [11,16,17].

The growth and technological development of the sector has affected the way farmers view the business. It has also affected the population of rural areas, who have seen new job opportunities. The nature of the tasks has changed, and most processes have ceased to be manual and started to become automated, which has in turn stimulated changes in the way human resources are managed in the Brazilian primary sector [18,19].

The sector presents particular characteristics that interfere with labor relations and with the implementation of HRM strategies by the industry and cane production, as well as other actors of the sector. Among them are the strong interference by the state and the need for rapid modernization due to recent deregulation of the sector, as well as technological changes that have transformed previously rudimentary and operational employee activities [18]. Thus, the post-burning cutting of sugarcane, which once involved a large number of workers, will offer fewer and fewer jobs. According to Carreira and Franco [19], mechanization in the sugar and alcohol sector has reduced the demand for low-skilled workers, expelling them from the field. The specialization of activity imposes the need for the qualification and training of a small group of workers to prepare them to meet the new requirements and challenges of the sugarcane operation.

Sugarcane labor is also changing because of technological advances that are generating a need for qualified people in leadership positions. According to Soares [20], due to the professionalization and consolidation of the sugar and alcohol industries, the demand has grown for middle- and upper-management executives in companies; this demand could be met by recruiting from among staff. However, for historical and cultural reasons, the surrounding localities often do not have skilled labor, thus forcing companies to seek professionals from other regions.

Albuquerque and Leite [21] warned of the cost and structure changes for companies related to the adaptation problems of people arriving from outside the region to work. According to them, this aspect still constitutes an imminent challenge for companies far from large population centers. In this context, HRM appears as an important actor as it is responsible for providing support to individuals involved in such a transformation, as well as providing greater effectiveness for their activities.

In these cases, academic studies have highlighted the need for motivational and behavioral programs. Leite et al. [22] pointed out, for example, that in remote areas, HRM should be linked essentially to the commitment of employees, especially in regions where there are units with remote organizational structures.

*2.2. HRM, Economic Development, and a Crossroad Between Traditional Methods and Modern Strategies*

Studies on HRM indicate that the effective use of the potential of employees provides a competitive advantage to an organization. Although it is difficult to measure the relationship between the HRM model(s) and organizational performance, several studies have described the variables that interact between these two components [23–26]. Modern HRM is intensely focused on the sustainability of the company's core issues and the alignment of departments to implement business strategies [23,24]. Other issues are in developing a holistic view based on the knowledge and involvement of other organizational subsystems [27], in the interlocution and conciliation of interests among different target audiences [21,28,29], and in securing adherence to the organizational changes required to implement the strategies [26,30].

In the debate about economic development and sustainable economic sectors, there is a trend toward the scope of strategic management (SM). Strategic management has focused on the contribution of HRM to organizational results, as a source of competitive advantages for companies based on the provision of qualified professionals in strategic areas of a business [25,26,28,31–34]. Another new HRM characteristic focuses on its strategic alignment and participation in organizational decisions as a source of positive results for companies. Albuquerque and Leite [33] argued that the control strategy, characterized by the non-durable exchange relationship between leaders and those being

led, has been replaced by the commitment strategy, characterized by the articulation of shared experiences and meanings that allow the work to be more participatory and to achieve better results, both individually and collectively, thereby establishing paths to growth and prosperity. HRM, for example, is nowadays responsible for developing a "holistic" view of employees from an understanding of organizational processes and for creating an integrated and cooperative working environment. The management of the organizational climate, the quality of life and safety at work, the social responsibility, and continuing development of employees are among the sustainability goals imposed by modern companies [29,35]. However, several studies point to the fact that some companies in certain sectors are still far from this strategic management of personnel [10,28]. Some of the causes for this distance from the strategic management of personnel are the economic culture in the sector, the level of employees' skills, the goals of companies' managers, or the foci of these managers [27,29,36].

To let economic development guide a sector's companies toward modern HRM, some barriers must be removed. Some of these barriers are related to the existence of internal conflicts between the chains of command in a business, resistance to change, and a limited vision of workers in their tasks, thereby impairing the perception of processes, as a whole as well as opportunities for growth and development [36]. In addition, some companies tend to adopt a more traditional management style, either because of their historical characteristics, their culture, their location, or their industry [32]. Thus, the second assumption of the research considers that the HRM model practiced in the sector and in the region is affected by the historical and cultural aspects typical of sugar–alcohol activities. The third assumption recognizes the contradiction between pressure for change and the traditional operational character of the sector, compelling HRM to adopt a strategic perspective [37]. The differences between traditional human resource management and strategic human resource management (SHRM) are summarized in Table 1.

**Table 1.** Differences between traditional Human Resource Management (HRM) and strategic HRM.

| Dimensions | Traditional HRM | Strategic HRM |
|---|---|---|
| Human Resources | Human resources are considered in functional roles | Human resource considerations as strategic planning |
| Employees | Employees are seen as factors in production | Employees are seen as assets |
| Communication | Top-down communication | Effective and open communication |
| Participation of Employees | Restricted participation and minimum involvement of employees | More participation and greater involvement of employees |
| Training and Career | Standard training and restricted career development opportunities | Better training and more career development opportunities |
| Rewards | Rewards according to task accomplishment | Rewards and recognition tied to performance |
| Premises | Constraints and tradition | Flexibility and innovation |

Sources: Jiang and Messersmith, 2018; Sammartino, 2002; Albuquerque, 2002; Boxall and Purcell, 2008; Henriques et al., 2012.

Therefore, we want to identify which model of HRM is followed by the current Brazilian sugar cane brandy companies. Are these companies following traditional methods of HRM ('resistant' to changes in the evolving economic development) or are they adopting modern profiles of HRM? Other questions emerging from our work relate to the adoption of policies toward a sustainable sector. What are the goals of these companies' managers in terms of economic sustainability? Subsequent sections will clarify the answers to these motivating questions.

## 3. Methodological Procedures

The research method had a qualitative approach, drawing from a descriptive mode, supported by a transversal field study. The first objective of this method was to describe the reality of HRM in sugar and alcohol industries, in their natural location, through on-site visits and interviews. Afterwards, we collected data and information from respondents from the agricultural companies. Finally, we did a detailed analysis of the collected data, considering the goal of observing which trend of HRM is dominant in the sector.

### 3.1. Research Population

This research focused on all 24 companies in the sugar and alcohol industries in the state of Mato Grosso do Sul, Brazil. Thus, the sample criterion was not adopted because we surveyed the total population of companies in the researched sector. In the case of companies with more than one operational unit in the state of Mato Grosso do Sul, only one unit was responsible for providing the information related to the group. This criterion reduced the number of respondents from 24 to 19 companies since there are two groups with three units each, and a third group that has two units in the state. To finalize the panel of participants, we had to exclude two companies which had ended their production activities in the previous year. Thus, 17 managers of these companies were surveyed. Of these 17 companies, nine are mechanized, three are partly mechanized, and five use manual harvesting. We consider as a mechanized company one in which the harvest is done by means of harvesting machinery. The semi-mechanized company is one in which the cutting or sorting of the harvest is done partly by human labor or by harvesting machines.

### 3.2. Data Collection and Data Analysis

Data collection was done through interviews, and for the primary analysis of the data, the technique of thematic analysis was adopted, as it is considered one of the most appropriate forms for qualitative investigations [38–41]. The purpose of the thematic analysis is to count the themes or items of meaning in a predetermined coding unit, allowing the quotations to be considered as units to be encoded [42]. In the next step, the categories of the analysis of the results were defined based on the HRM dimensions described in the theoretical review. We considered the following categories: work relations, employee recruitment and selection, training and development, compensation and benefits, occupational safety and health, quality of life at work, and the main challenges. Table 2 summarizes these differences.

**Table 2.** HRM policies and practices (literature summary).

| HRM Policies | HR Practices | |
|---|---|---|
| | **Traditional View** | **Strategic View** |
| **Recruitment and selection** | Local recruitment based on case-by-case applications | Planning of recruitment and selection activities according to the demands of various sectors and the company's strategic guidelines |
| **Training and development** | Lack of structured plans for professional development | Transcendence of compulsory training and greater integration between HRM and managers of various areas in order to diagnose needs |
| | | Possibility of professional development within the organization |
| | Professional development focused on the operational objectives of the company | Preference for the application of training involving human skills, in addition to techniques |
| | | A focus on the development and ongoing education of employees |
| **Wages and benefits** | Fixed remuneration per contract without incentive systems for productivity, development, merit, or by social extension | Systems of remuneration for knowledge, bonuses for performance, and evaluation of competencies of employees |
| **Relationships between the company and workers/employees** | Relationships supported by management of variable costs to the company | Relationships for valuing opinions and ideas on how to improve results |
| | Lack of commitment to create additional links between the company and the employee | Encouragement of socialization that strengthens workers' identification with the company |
| | | Internal communication programs in the company Strengthening labor relations, leading to greater commitment |
| **Working conditions and quality of life** | Risk management without contingency planning Maintenance of equipment and materials up to the maximum useful life | Existence of internal commission for accident prevention and risk management |
| | Absence of events involving employees' families | Dissemination of results and events involving the company, leading to greater transparency; Access to adequate equipment and materials; Hospital health care; Health plans or private pension plans; Events for employees and their families |

Source: Adapted from Fiuza (2008); Albuquerque and Leite (2009); Leite et al. (2010).

The policies and practices presented in Table 2 are a systematization of the characteristics of the new role played by HRM in companies. Thus, it becomes possible to further verify whether the investigated companies are carrying out these policies and practices (if they are 'innovative' companies) or if they are still concentrating their efforts on the more traditional functions of HRM (i.e., whether they are 'change resistant').

*3.3. Presentation of Research Findings*

Regarding the division of categories and their respective practices, a review of the literature on HRM in organizations was done. This review usually delimits five relevant themes, such as (i) work relationships, (ii) recruitment and selection, (iii) training and development, (iv) remuneration and (other) benefits, and (v) occupational health and safety. The theme of work relationships includes valuing the opinions of employees, socialization, and identification with the company, and also considers internal communication programs and commitment. The recruitment and selection category contains the planning of recruitment and selection activities according to the demand of the various sectors and the company's strategic guidelines. The training and development category involves the integration between HRM and the managers of the departments to diagnose the organization's training and skills needs, the professional development, and the continuous education of employees. The remuneration and benefits category includes the compensation and profit-sharing program, and the compensation criteria based on knowledge and performance evaluation of employees. Finally, the category entitled occupational Health and safety includes accident prevention, risk management, access to appropriate equipment and materials, medical assistance, health plans, and events for employees and their families. Therefore, the characteristics identified in the companies were categorized according to these parameters of analysis.

Category 1—Work Relationships (RT)

Initially, the companies were divided into two groups: (A) companies with low mechanization in the production of sugarcane and (B) companies with mechanized production.

In Group A, where planting occupies large areas, the harvest is done manually. The links between the company and its employees are limited to legal aspects. No effort is made to create lasting bonds with workers. HRM offers only temporary support to workers, and there are no actions taken to commit and retain qualified people. The contracts are for a fixed term and end at the completion of the harvest.

In Group B, companies with mechanized processes retain a small number of rural workers (about 17%) to cover areas where the use of machines is not possible. There is a preference for hiring local labor. Thus, it is possible to develop closer ties with employees and to strengthen working relationships and commitment policies, leading to a differentiated view of employees. In addition, due to the lack of skilled labor in the sector and the introduction of new technologies for soil preparation, planting, and harvesting, the need for professional qualification is greater. In this case, the retention of skilled employees is a priority, and termination at the end of the annual harvest becomes unnecessary [19,20]. The production period runs from February to December, with less than two months between the period for the layoff of rural workers and their hiring for the next year.

The practice of "favoring socialization to strengthen employee identification with the company" was identified in 12 interviews. These practices involve sponsoring company events and providing recreational areas within the company, such as programs for family visits.

"Internal communication programs in the company" were cited in nine cases; among them were incentives for communication between hierarchical levels of the company, sharing experience between departments, and direct channels for questions and suggestions. Another practice, that of "strengthening labor relations in the face of greater commitment," was identified in only six cases.

Finally, "valuing opinions and findings on how to obtain results" was mentioned by only three respondents. This small number indicates companies' limited efforts to listen to their employees and to value their opinions with regard to improvements in company processes.

In summary, in mechanized companies (Group B), labor relations tend to be more solid, starting with a greater effort to involve and retain people, and are integrated through socialization and internal communication. Group B is therefore akin to innovative companies in terms of HRM. In Group A (mostly composed of companies with little or no mechanization), labor relations are still constrained by the seasonality of workers and the high turnover of industrial and administrative employees. Labor relations in Group A companies indicate low employee participation, little communication, and exceptional employee retention. Therefore, in Group A dominates the group of change-resistant units.

Category 2—Recruitment and Selection (R&S)

R&S policies involve the processes of attracting, recruiting, and hiring people. In the case of sugar–alcohol companies, it is observed that the planning of personnel follows the demands of departments and the strategic guidelines of the company.

Although the procedures are similar, the use of recruitment techniques is differentiated between mechanized and non-mechanized companies. While mechanized subsectors prefer local rural workers, in the non-mechanized companies, the search is conducted in other states and usually brokered by autonomous professionals based in the states of preference, in this case, Minas Gerais, Alagoas, and Bahia. In the case of the recruitment and selection of workers in industrial and administrative areas, procedures are similar in both groups.

Among the main procedures, the following observations were collected:

(a) The methods for recruiting people are not standardized to meet immediate needs. No defined strategies were identified for this activity;

(b) Testing and interview scripts are not strategically defined. All respondents are led to selection through non-standardized approaches, sometimes through psychological tests (4) or through interviews (13). There is no clear definition of selection strategies;

(c) It is up to HRM to identify the profile and skills required and select potential candidates. The process is conducted jointly with the managers of the departments and directors of the company (4); and

(d) The connection between HRM and other departments, as well as the management of the company, is also intense in relation to the negotiation of vacancies (4).

In relation to the power to make decisions, there were two groups—the first with greater autonomy for decision-making and strategic participation, and the second with less autonomy, subject to the deliberations of the company's board of directors. Of the actors investigated, 23.5% (four cases) fall into the higher-autonomy group, and 76.5% (13 cases) had lower autonomy, reflecting the distance between HRM and the company's strategic decision-makers.

Still, in the group with less autonomy, there seemed to be no stimulus for contributing to changes and to the development of organizational flexibility. Although HRM is more integrated and works together with the departments, the data indicate that this action is restricted to transmitting executive deliberations to employees. For 76.5% of the cases, there is no evidence that HRM plays a strategic and transformative role, which is in close proximity to the conclusions of Fischer [43] and Jiang and Messersmith [37].

Category 3—Training and Development Programs (T&D)

This category refers to the presence of training and development programs. It considers the effort to improve the performance of employees, be it technical, disciplinary, or behavioral.

The lack of skilled labor in the sector justifies the maintenance of T&D policies and practices. All respondents stated that HRM has sought to work with managers to qualify employees. Of the 17 companies, only eight offer technical training; in three of these, there is a predominance of technical training and, possibly, of behavioral and disciplinary training. Only in six cases was there concern about training involving human skills, in addition to techniques. Few companies invest in the development of leadership skills from the "coordinator" level. Regarding the areas to which the employees involved in the above-mentioned training belong, only three companies include agricultural leaders; the others include only strategic and higher-level positions belonging to the industrial or administrative areas. In addition, it should be noted that, among the 17 companies, 14 do not seem to have a structured and regular career plan, according to the interviewees.

Category 4—Remuneration and Benefits Systems (R&B)

The remuneration and benefits to workers of the investigated companies were based on two parameters of analysis [24]: the first states that HRM offers structured remuneration programs with variables of participation in the results, and the second is based on remuneration for knowledge, performance bonuses, and skills assessment.

The responses in this category allowed us to verify that, with the exception of one company, 16 companies have relatively homogeneous practices. Remuneration is based on fixed salaries and some additional factors, with differences in concessions between the positions of the companies, according to whether they are rural, industrial, administrative, or organizational leadership positions.

For companies that employ rural workers to plant and cut sugarcane, the practice of rewarding additional productivity is common. In this system, workers receive variable remuneration, calculated according to the accomplishment of the goals of the team, which is formed by effective rural workers.

The employees in production and administrative areas do not receive remuneration based on additional productivity gains. However, in six cases, companies practice other forms of structured remuneration and offer profit sharing. There were no indications of remuneration for knowledge, bonuses for performance, or evaluation of competencies, which are summarized in the second parameter of analysis in this category.

The most common benefits in sugarcane companies cited by respondents are presented in Table 3.

Table 3 indicates compliance with legal obligations and other initiatives to support rural workers. Due to the characteristics of the work, the provision of canteens and transportation of workers to productive units are important benefits due to the distance between work and cities and for the viability of the operation.

**Table 3.** Benefits granted by Mato Grosso do Sul's sugar and alcohol companies.

| Benefits | No. of Cases |
|---|---|
| Health plan | 58% (10 cases) |
| Accommodation for migrant rural workers from other states | 29% (5 cases) |
| Transport of employees from nearby cities to productive units | 100% (17 cases) |
| Restaurant for employees of industrial and administrative areas | 100% (17 cases) |
| Traveling canteen to meet rural employees at their areas of planting or harvest, with water, space, and shade for resting, and tables for food | 47.05% (8 cases) |
| Leisure area for employees | 41.17% (7 cases) |

Source: own research data.

Category 5—Occupational Health and Safety (OHS)

OHS policies pay particular attention to reducing unsafe working conditions, which is a priority for all. In this category, the first parameter of analysis refers to the internal committee, which aims to raise awareness among employees about the safety and supervision of working conditions. It was

observed that all respondents recognize HRM as an important actor in creating awareness about employees' behavior changes, aiming at safe working conditions.

The second parameter of analysis in this category is the search for greater transparency in the disclosure of results and occurrences. A total of eight cases, or 47%, indicate the accomplishment of practices consistent with this parameter. In almost all companies, HRM is responsible for the issuance of policies and practices for occupational safety and accident prevention in production units.

### 3.4. Discussion of Results

The data obtained reveal that the sugar and alcohol companies of the Mato Grosso do Sul (MS) have different characteristics in each category. This reflects the different reactions to socio-economic development stimuli and to heterogeneous sustainability requirements.

In category 1, it was observed that technological aspects have great weight in the model of the performance of HRM due to the profile of the workforce. Thus, this study corroborates the view of Carreira and Franco [19] that the mechanization of sugarcane production has changed the way of dealing with workers in the mills since it has reduced the demand for less-qualified workers and increased the need for employee training.

Mechanized companies experience a new form of labor relations that allows them to build links with the employees. With this, HRM starts demanding greater integration. In this case, higher priority is given to the retention of workers, especially in the hiring of more qualified local employees, thereby reducing the hiring of migrant workers who depend on the company for housing and food. These results support the considerations of Albuquerque and Leite [33] regarding the problems of workers from outside the region and the higher costs of infrastructure. Additionally, these results reinforce the idea that the more mechanized companies are also more able to follow strategic management guidelines for HRM.

In category 2 (R&S), it was verified that policies and practices, as well as being linked to labor relations, are differentiated by the autonomy of HRM. In most companies, structured R&S policies and practices were found along with intense relationships with departments. In the selective processes, variability was found in the form of evaluations and interviews, and strategic positions are passed by the managers of the departments for which they are intended. Despite the interaction between HRM and these departments, the function remains operational and informational.

Another outstanding issue in this category is the role of management in negotiating vacancies and departmental needs, positioning HRM as an internal issue. These results are aligned with the proposals of Sammartino [44], who reported the participation of HRM in the role of consultant in the various areas of the organization. According to the author, this action demands a greater strategic and generalist performance of HRM to recognize how the use of the organization's human capital can be improved.

An effort was made by HRM to act more proactively and closely with the internal customer. However, many companies do not give HRM the autonomy to do so. In 13 of the 17 cases examined, HRM is a control entity, responsible for providing indicators to the board according to predetermined guidelines. This reality was supported by Silva et al. [45], who pointed out a contradiction between the need to reconcile various organizational interests and the daily reality.

In category 3 (T&D), given the lack of skilled workers and the difficulty of bringing them in from other locations, efforts were made to develop employees beyond legal aspects. It is posited that the ideal situation would be to develop skills and offer continuous training, aiming to meet the typical needs of companies away from large centers [19,21]. It was noted that there is a preference for technical training and that continuous training programs are the exception.

With regard to category 4 (R&B), attracting workers, valuing talent, and maintaining high-potential and qualified personnel are still distant realities for the Brazilian sector since most companies report that policies and practices are focused on immediate outcomes, to the detriment of the long term outcomes. There do not seem to be career policies in these organizations. However, practices aimed

at motivating and mobilizing groups toward organizational goals have been identified, which is also a recommendation of Leite et al. [21].

In category 5, the results indicate that HRM is directly linked to safety and to the prevention of accidents in the work environment. Because of the risks, a set of initiatives supported by the law have been undertaken. These measures include the protection of employees, the existence of an internal commission to prevent accidents, and the intensive participation of HRM in the awareness of employees [27,35,46,47].

The challenges experienced by HRM are conditioned by internal barriers. Among them, we must note the lack of autonomy to solve problems, the difficulty of integration among departments, and the existence of divergent interests for the strategic action of the area. These obstacles reinforce the traditional character of HRM in the sugar and alcohol industry, justifying the adoption of more operational and standardized policies and practices, albeit with differences.

In summary, in categories 1, 2, 3, and 4, the traditional model prevailed. These companies tend to be resistant in these categories. The investments made by the large sugar–alcohol groups increase pressure for the modernization of the sector and, consequently, increase demand for skilled labor, and increase responsibilities conferred on HRM, as explained by the studies of the sector [18,19]. Despite this, there are changes.

A total of nine companies adopted stricter result controls and fewer ties with workers, while the other eight cases invested in the development of human capital, supporting the view of Lacombe and Chu [48], according to whom "companies of the same sector and with similar products can adopt different strategies and configurations for the HRM."

It was also observed that the characteristics of the companies investigated vary between traditional ('resistant') and strategic ('modern') in the different categories. That is, the same company can present policies and strategic practices for training and developing people but archaic policies and practices for compensation and benefits. This type of simultaneity has been found to be a relevant achievement in our discussion, revealing the transition path characterizing the studied companies in terms of economic development.

The research findings showed that HRM plays an integrating role between the other departments and the management of the companies. HRM's contribution to other departments was also observed, mainly in category 4 (training and development) and category 5 (health and safety at work), as the results indicate that it is increasingly up to HRM to present solutions for the contingencies of the organizational daily life. However, it is observed that HRM transmits and conveys the company's strategy, but does not have a constant participation in strategic definitions. In only a few cases, HRM participates in decision-making related to category 2 (recruitment and selection) and category 4 (remuneration and benefits). In general, there is a lack of autonomy in the HRM area, a fact that impels it to adopt a standardized and predetermined behavior in the face of the problems that occur. Thus, there is a paradox as to the characteristics of HRM in sugar and alcohol companies, because even in the search for competitive advantages obtained by the better use of people, its policies and practices still seem to be plastered and pre-determined, without opening for questioning and without overcoming existing challenges. Therefore, HRM of the sugar and alcohol sector in this research findings tends to assume a traditional and slightly strategic role in its subsystems, which are labor relations, recruitment and selection, training and development, and remuneration and benefits. Only for the occupational health and safety category is there a proactive role for HRM, with greater participation in the design of programs and activities focused on the well-being of employees. Therefore, the results indicate that the traditional HRM precepts prevail, culminating in an operational focus as seen in categories one (1), two (2), three (3), and four (4) of the data.

Table 4 presents the predominant characteristics.

**Table 4.** Predominant characteristics in relation to the analyzed categories.

| Categories | Analyzed Parameters | Dominant Characteristics | Evidence of Preferred HRM Model |
|---|---|---|---|
| Recruitment and Selection (R&S) | Integrated R&S planning according to sector demand and strategic company guidelines | R&S structured on demand in strategic areas and guidelines Variability in the selection process, interviews, and tests Tendency for closer relations with other departments | Traditional |
| Training and Development (T&D) | Possibility of professional development within the organization Preference for training in human skills, in addition to techniques Focus on the development and continuing education of employees | Lack of structured plans for professional development Preference for technical training based on industrial and agricultural needs Search for development and education of employees, in well-defined situations | Traditional |
| Remuneration and Benefits | Structured remuneration program with variables based on profit sharing Systems of remuneration for knowledge, bonuses for performance, and evaluation of employees' competencies | Structured remuneration program based on fixed and additional productivity wages No remuneration for knowledge, bonuses for performance, or evaluation of competencies | Traditional |
| Relationship between Company and Employees | Valuing opinions and discoveries about how to obtain results Strengthening the workers' identification with the company Internal communication programs in the company Strengthening labor relations, aiming at greater commitment | Relations are longer lasting and narrower Focus on the retention of qualified people Priority given to local resident workers Encouragement of socialization to strengthen relationships | Traditional |
| Occupational Health and Safety | Existence of Internal Commission of Accident Prevention (CIPA) and risk management Dissemination of data and facts, aiming for transparency Access to adequate equipment and materials Hospital medical care Health plans and/or private pension plans Events for workers and their families | Existence of CIPA and risk management Dissemination of data and facts about the company, aiming at transparency Access to adequate equipment and materials Hospital medical care Adoption of health plans by some companies and absence of private pension programs Events for employees and their families | Strategic |

Source: own work.

## 4. Conclusions

This article aimed to identify the dominant profile of the management of human resources in the Brazilian sugar and alcohol sector, considering socio-economic development and challenges imposed on the companies of the sector.

The first challenge imposed by economic development on the HRM of this sector regarded the evolution of the traditional HRM model to a more modern/strategic model, induced by pressures derived from technology, globalization, and competitiveness. In fact, the pressure for change is clear, as modern agriculture has brought forward a set of technological standards that have culminated in increased productivity and competitiveness. It is also true that the volume of exports has increased, mainly due to the production of alcohol and ethanol. However, evolution is most clearly noticeable in productive environments that use high-technology resources the most. The differences between HRM models have been basically found to be related to the level of applied technology and, consequently, to the qualification of the workforce.

The second examined challenge evaluates that the HRM model practiced in the sector is affected by the historical–cultural aspects typical of the region. It can be affirmed that the distance from Brazil's large industrial centers and the low number of schools to qualify the labor force, in addition

to topographic aspects, have limited the possibilities of using machines for planting and harvesting. In addition, one must take into consideration the cultural pattern of leaders, which impacts to greater or lesser resistance on change. This observation is directly related to the third assumption that examines the contradiction between pressure for change and the traditional operational character of the sector, compelling HRM to adopt a strategic perspective. Regarding this possibility, it was observed that, in all the companies, the environment allows for combining HRM's actions with operational and strategic activities, establishing an intermediate level of performance.

Thus, more than a contradiction, there is a challenge in HRM of sugar and ethanol companies. Although some companies seek competitive advantages for the better use of employee skills, policies and practices are standardized, making it difficult to overcome the challenges in the sector. Therefore, responding to the research question, HRM of the sugar and alcohol sector of the state of Mato Grosso do Sul is still predominantly traditional. As a consequence, we claim that any planning for the socio-economic development of the region must recognize this 'resistant' dimension and enhance proper methods for combining this resistance with the need for modernization of the sector and with the need for innovation in HRM practices.

As a contribution of this research, the findings that characterize HRM in agricultural organizations are highlighted, inserted into a context of concerns for sustainability and social responsibility, in the Brazilian scope. It was possible to portray HRM in a detailed way in view of the scarce research on the topic. This research also contributed to the incentive for studies on remote and peculiar sectors, marginalized from large centers and facing greater resource scarcity and unique challenges, such as the sugar and alcohol sector, which was established in the vast plains, previously dedicated only to grain planting and livestock.

As challenges, we intend to develop robust content analysis of the provided responses by the interviewed managers, namely by using AlCeste methods. We also intend in the next derivations of this work to extend the discussion of the results, supported by the use of indicators that could provide evidence for the studied dimensions, giving scope to examine eventual causal relationships or even temporal perspectives. Additionally, we intend to extend the methodology used here to other spaces in the federation of Brazilian in order to verify the predominance of traditional human resource management in the diversity of primary sectors of the Brazilian economy.

**Author Contributions:** P.M.: substantial contributions to conception and design and analysis and interpretation of data; E.K.: substantial contributions to conception and design and analysis and interpretation of data; I.S.: substantial contribution to interpretation of data and participation in drafting the article and revising it critically for important intellectual content; V.M.: substantial contribution in acquisition of data and analysis and interpretation of data. All authors have read and agreed to the published version of the manuscript.

**Funding:** This research received no external funding.

**Conflicts of Interest:** The authors declare no conflict of interest.

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
