# Peer review of "Economic Development and Changes in Human Resource Management in a Sustainable Agricultural Sector: Recent Evidence from Brazilian Sugar–Alcohol Companies"

_sustainability, doi:10.3390/su12187559_

Round 1

Reviewer 1 Report

I think that this paper deals with an interesting topic, following an appropriate approach, both in theorical and empirical analysis. It is needed to analyse the role of human resources in traditional sectors like agriculture. This sector has changed in the last decades due to an increasing mechanization process. It challenges the idea of a lagging sector or economy, as well as the low-skilled workers. In this sense, the role of the agents in that process is critical for innovation and sustainability.

The abstract is complete, including the relevant information. The topic is well contextualized and poses clearly the research aims. The work is well structured.

The theoretical approach is appropriate (growth processes Perroux and the different responses to the pace of economic development). In particular, the paper focuses on the sustainability of resources management in sugar-alcohol companies in Brazil and specifically it analyses if the sector is dominated by promoter of changes or by resistant to changes. It is good the connection between sustainability, development with the profile of economics actors. This statement can also be applied to innovation.

I appreciate the methodological approach based on qualitative research. I appreciate the information provided in tables 2 and 3.

The conclusions are founded in the research and limitations and future extensions are also included.

I consider that the paper can be published. However, I suggest some minor changes.

  • I consider that is needed to define mechanized companies. It is more necessary because the paper set two types in the category 1 (group A and group B). I think that it could be interesting to have three groups according to the different level of mechanization. If I had the definition of each group, I can value better the results and if it is needed a new group or not.
  • I suggest highlighting the own contribution in the introduction and/or conclusions of the paper.
  • I suggest describing better the objectives and/or contents of the interviews. One or two sentences can help to a better understanding.
  • Page 1, the reference Pereira et al. should include the year.
  • I suggested avoiding use abbreviations in the abstract
  • Page 2, lines 70-75. The explanation between brackets (last sentence of the paragraph) can be included in the text.
  • Section 2.3 can be merged some paragraphs clearly connected. The same in section 3.1.
  • The legend of figure 1 is not in English.
  • The references should be ordered. Some references should also be checked to follow the same style in the order of the year and in putting the name or only the initial of the authors. For instance, in the reference Guessart the year is at the end.
  • Finally, I suggest polishing up the English. The writing is fluid and easy-reading, but there are some minor errors (for instance, innovational is not usual, I suggest innovative) or even some sentence without clear subject.

Author Response

Dear Reviewer 1,

We thank you for your deep attention on the previous version of this research. As fully deserved, we inserted an initial acknowledgement recognizing Sustainability’s reviewers. Let us also highlight how we became highly stimulated to produce this revision after reading your so supportive sentences: “I think that this paper deals with an interesting topic, following an appropriate approach, both in theorical and empirical analysis. It is needed to analyse the role of human resources in traditional sectors like agriculture. This sector has changed in the last decades due to an increasing mechanization process. It challenges the idea of a lagging sector or economy, as well as the low-skilled workers. In this sense, the role of the agents in that process is critical for innovation and sustainability. // The abstract is complete, including the relevant information. The topic is well contextualized and poses clearly the research aims. The work is well structured. // The theoretical approach is appropriate (growth processes Perroux and the different responses to the pace of economic development). In particular, the paper focuses on the sustainability of resources management in sugar-alcohol companies in Brazil and specifically it analyses if the sector is dominated by promoter of changes or by resistant to changes. It is good the connection between sustainability, development with the profile of economics actors. This statement can also be applied to innovation. // I appreciate the methodological approach based on qualitative research. I appreciate the information provided in tables 2 and 3. // The conclusions are founded in the research and limitations and future extensions are also included.”

Therefore, let us detail the main changes of this version:

  • The title has been slightly changed.
  • English language has been polished.
  • We have not used abbreviations in the Abstract.
  • Now, you can find a clear proposition of the general objective of the research as well as its specific objectives: “The general objective of this research is therefore to evaluate how the processes of economic development have inserted changes in the HRM performed in sugar and alcohol companies in Brazil. As specific objectives, which will allow the achievement of the general objective of the research, the following stand out: // • To discriminate the main impacted fields of action of the HRM, and, therefore, to establish items to verify their characteristics; • To check in the field (i.e., in the sugar and alcohol companies of Brazil), how the stimuli from the processes of economic development have led to significant modifications (or not) in the characteristics studied by the HRM.”
  • Former sections 2.1 and 2.2 have been synthesized.
  • Especially along the new Section 3 we detail the research method used in this paper.
  • Former Section 3-1’s paragraphs have been merged.
  • Now, we clearly defined “Mechanized Company”. We also define semi-mechanized company: “We consider as a mechanized company the one in which the harvest is done by means of harvesting machinery. The semi-mechanized company is one in which the cutting or sorting of the harvest is done partly by human labor or by harvesting machines.”
  • We clarify that our Data Analysis is explicitly inserted in Section 3.2.
  • Now, we also detailed the construction of the categories used in the Analysis: “Regarding the division of categories and their respective practices, a review of the literature on HRM in organizations was previously done. This review usually delimits five relevant themes, such as (i) Labor Relations, (ii) Recruitment and selection, (iii) Training and development, (iv) Remuneration and (other) benefits and (v) Occupational Health and Safety. Labor relations include valuing the opinions of employees, socialization and identification with the company, also considering internal communication programs and commitment. The Recruitment and Selection category contains the planning of recruitment and selection activities according to the demand of the various sectors and the company's strategic guidelines. The Training and Development category involves the integration between HRM and the managers of the areas to diagnose the organization's training and skills needs, the professional development and continuous education of employees. The Remuneration and benefits category includes the compensation and profit sharing program, the compensation criteria based on knowledge and performance evaluation of employees. Finally, the category entitled Occupational Healthy and Safety includes accident prevention, risk management, access to appropriate equipment and materials, medical assistance, health plans and events for employees and their families.”
  • We synthesize now the major achievements upon our discussion: “It was also observed that the characteristics of the companies investigated vary between traditional (‘resistant’) and strategic (‘modern’) in the different categories. That is, the same company can present policies and strategic practices for training and developing people but archaic policies and practices for compensation and benefits. This type of simultaneity has been found a relevant achievement in our discussion, revealing the transition path characterizing the studied companies in terms of economic development.
  • Table 4 presents the predominant characteristics.
  • [Insert here Table 4]
  • The research findings showed that HRM plays an integrating role between the other departments and the management of the companies. HRM's contribution to other departments was also observed, mainly in category 4 (Training and Development) and category 5 (Safety and Health at work), as the results indicate that it is increasingly up to HRM to present solutions for the contingencies of the organizational daily life. However, it is observed that HRM transmits and conveys the company's strategy, but does not have a constant participation in strategic definitions. In only a few cases, HRM participates in decision making related to categories 2 (Recruitment and selection), category 4 (Remuneration and Benefits). In general, there is a lack of autonomy in the HRM area, a fact that impels it to a standardized and predetermined behavior in the face of the problems that occur. Thus, there is a paradox as to the characteristics of HRM in sugar and alcohol companies, because even in search of competitive advantages obtained by better use of people, its policies and practices still seem to be plastered and pre-determined, without opening for questioning and without achieving overcome existing challenges. Therefore, the HRM of the sugar and alcohol sector in this research findings tends to assume a traditional and little strategic role in its subsystems, which are labor relations, recruitment and selection, training and development and remuneration and benefits. Only for the Occupational Health and Safety category and observes a proactive role for HRM, with greater participation in the design of programs and activities focused on the well-being of employees. Therefore, the results indicate that the traditional HRM precepts prevail, culminating in an operational focus as seen in categories one (1), two (2), three (3), and four (4) of the data.”
  • As a contribution of this research, the findings that characterize HRM in agricultural organizations are highlighted now in the Conclusions.

After this extensive effort, we thank you again for your attention. Now, we are aware that we are submitting a much more appropriate piece to be published by Sustainability.

Yours,

The Authors.

Reviewer 2 Report

Research objectives are not clearly disclosed. What are the aims of this paper?

Research methods and data analysis are missing.

Author Response

Dear Reviewer 2,

We thank you for your deep attention on the previous version of this research. As fully deserved, we inserted an initial acknowledgement recognizing Sustainability’s reviewers.

Therefore, let us detail the main changes of this version:

  • The title has been slightly changed.
  • English language has been polished.
  • We have not used abbreviations in the Abstract.
  • Now, you can find a clear proposition of the general objective of the research as well as its specific objectives: “The general objective of this research is therefore to evaluate how the processes of economic development have inserted changes in the HRM performed in sugar and alcohol companies in Brazil. As specific objectives, which will allow the achievement of the general objective of the research, the following stand out: // • To discriminate the main impacted fields of action of the HRM, and, therefore, to establish items to verify their characteristics; • To check in the field (i.e., in the sugar and alcohol companies of Brazil), how the stimuli from the processes of economic development have led to significant modifications (or not) in the characteristics studied by the HRM.”
  • Former sections 2.1 and 2.2 have been synthesized.
  • Especially along the new Section 3 we detail the research method used in this paper.
  • Former Section 3-1’s paragraphs have been merged.
  • Now, we clearly defined “Mechanized Company”. We also define semi-mechanized company: “We consider as a mechanized company the one in which the harvest is done by means of harvesting machinery. The semi-mechanized company is one in which the cutting or sorting of the harvest is done partly by human labor or by harvesting machines.”
  • We clarify that our Data Analysis is explicitly inserted in Section 3.2.
  • Now, we also detailed the construction of the categories used in the Analysis: “Regarding the division of categories and their respective practices, a review of the literature on HRM in organizations was previously done. This review usually delimits five relevant themes, such as (i) Labor Relations, (ii) Recruitment and selection, (iii) Training and development, (iv) Remuneration and (other) benefits and (v) Occupational Health and Safety. Labor relations include valuing the opinions of employees, socialization and identification with the company, also considering internal communication programs and commitment. The Recruitment and Selection category contains the planning of recruitment and selection activities according to the demand of the various sectors and the company's strategic guidelines. The Training and Development category involves the integration between HRM and the managers of the areas to diagnose the organization's training and skills needs, the professional development and continuous education of employees. The Remuneration and benefits category includes the compensation and profit sharing program, the compensation criteria based on knowledge and performance evaluation of employees. Finally, the category entitled Occupational Healthy and Safety includes accident prevention, risk management, access to appropriate equipment and materials, medical assistance, health plans and events for employees and their families.”
  • We synthesize now the major achievements upon our discussion: “It was also observed that the characteristics of the companies investigated vary between traditional (‘resistant’) and strategic (‘modern’) in the different categories. That is, the same company can present policies and strategic practices for training and developing people but archaic policies and practices for compensation and benefits. This type of simultaneity has been found a relevant achievement in our discussion, revealing the transition path characterizing the studied companies in terms of economic development.
  • Table 4 presents the predominant characteristics.
  • [Insert here Table 4]
  • The research findings showed that HRM plays an integrating role between the other departments and the management of the companies. HRM's contribution to other departments was also observed, mainly in category 4 (Training and Development) and category 5 (Safety and Health at work), as the results indicate that it is increasingly up to HRM to present solutions for the contingencies of the organizational daily life. However, it is observed that HRM transmits and conveys the company's strategy, but does not have a constant participation in strategic definitions. In only a few cases, HRM participates in decision making related to categories 2 (Recruitment and selection), category 4 (Remuneration and Benefits). In general, there is a lack of autonomy in the HRM area, a fact that impels it to a standardized and predetermined behavior in the face of the problems that occur. Thus, there is a paradox as to the characteristics of HRM in sugar and alcohol companies, because even in search of competitive advantages obtained by better use of people, its policies and practices still seem to be plastered and pre-determined, without opening for questioning and without achieving overcome existing challenges. Therefore, the HRM of the sugar and alcohol sector in this research findings tends to assume a traditional and little strategic role in its subsystems, which are labor relations, recruitment and selection, training and development and remuneration and benefits. Only for the Occupational Health and Safety category and observes a proactive role for HRM, with greater participation in the design of programs and activities focused on the well-being of employees. Therefore, the results indicate that the traditional HRM precepts prevail, culminating in an operational focus as seen in categories one (1), two (2), three (3), and four (4) of the data.”
  • As a contribution of this research, the findings that characterize HRM in agricultural organizations are highlighted now in the Conclusions.

After this extensive effort, we thank you again for your attention. Now, we are aware that we are submitting a much more appropriate piece to be published by Sustainability.

Yours,

The Authors.

Reviewer 3 Report

Dear Author(s),

Thank you for your submission.

This paper examines Economic Development and Sustainability in the Agricultural Sector. The paper, generally, is well organized and presents an interesting and robust empirical part. My feedback suggests specific areas in which you can further strengthen your current submission.

  1. I feel that the title is not completely adequate, maybe authors can be more specific in the title
  2. I consider that the literature review is too long, maybe sections 2.2 and 2.1 can be reorganized and be together, in a shorter format
  3. It is not completely clear how authors divide and organize the categories presented in section 3.3 maybe they can add some extra information about what is included in each category
  4. We miss a deeper discussion in section 3.4. I think you should reinforce and explain each of the categories and explain their main implications. Maybe you can add a final paragraph presenting and explaining the most important results and its impacts, has a complement of table 5, that is quite interesting.

This concludes my feedback. I hope it might be helpful.

Author Response

Dear Reviewer 3,

We thank you for your deep attention on the previous version of this research. As fully deserved, we inserted an initial acknowledgement recognizing Sustainability’s reviewers. Let us also highlight how we became highly stimulated to produce this revision after reading your so supportive sentences: “This paper examines Economic Development and Sustainability in the Agricultural Sector. The paper, generally, is well organized and presents an interesting and robust empirical part.”

Therefore, let us detail the main changes of this version:

  • The title has been slightly changed.
  • English language has been polished.
  • We have not used abbreviations in the Abstract.
  • Now, you can find a clear proposition of the general objective of the research as well as its specific objectives: “The general objective of this research is therefore to evaluate how the processes of economic development have inserted changes in the HRM performed in sugar and alcohol companies in Brazil. As specific objectives, which will allow the achievement of the general objective of the research, the following stand out: // • To discriminate the main impacted fields of action of the HRM, and, therefore, to establish items to verify their characteristics; • To check in the field (i.e., in the sugar and alcohol companies of Brazil), how the stimuli from the processes of economic development have led to significant modifications (or not) in the characteristics studied by the HRM.”
  • Former sections 2.1 and 2.2 have been synthesized.
  • Especially along the new Section 3 we detail the research method used in this paper.
  • Former Section 3-1’s paragraphs have been merged.
  • Now, we clearly defined “Mechanized Company”. We also define semi-mechanized company: “We consider as a mechanized company the one in which the harvest is done by means of harvesting machinery. The semi-mechanized company is one in which the cutting or sorting of the harvest is done partly by human labor or by harvesting machines.”
  • We clarify that our Data Analysis is explicitly inserted in Section 3.2.
  • Now, we also detailed the construction of the categories used in the Analysis: “Regarding the division of categories and their respective practices, a review of the literature on HRM in organizations was previously done. This review usually delimits five relevant themes, such as (i) Labor Relations, (ii) Recruitment and selection, (iii) Training and development, (iv) Remuneration and (other) benefits and (v) Occupational Health and Safety. Labor relations include valuing the opinions of employees, socialization and identification with the company, also considering internal communication programs and commitment. The Recruitment and Selection category contains the planning of recruitment and selection activities according to the demand of the various sectors and the company's strategic guidelines. The Training and Development category involves the integration between HRM and the managers of the areas to diagnose the organization's training and skills needs, the professional development and continuous education of employees. The Remuneration and benefits category includes the compensation and profit sharing program, the compensation criteria based on knowledge and performance evaluation of employees. Finally, the category entitled Occupational Healthy and Safety includes accident prevention, risk management, access to appropriate equipment and materials, medical assistance, health plans and events for employees and their families.”
  • We synthesize now the major achievements upon our discussion: “It was also observed that the characteristics of the companies investigated vary between traditional (‘resistant’) and strategic (‘modern’) in the different categories. That is, the same company can present policies and strategic practices for training and developing people but archaic policies and practices for compensation and benefits. This type of simultaneity has been found a relevant achievement in our discussion, revealing the transition path characterizing the studied companies in terms of economic development.// Table 4 presents the predominant characteristics. // [Insert here Table 4]// The research findings showed that HRM plays an integrating role between the other departments and the management of the companies. HRM's contribution to other departments was also observed, mainly in category 4 (Training and Development) and category 5 (Safety and Health at work), as the results indicate that it is increasingly up to HRM to present solutions for the contingencies of the organizational daily life. However, it is observed that HRM transmits and conveys the company's strategy, but does not have a constant participation in strategic definitions. In only a few cases, HRM participates in decision making related to categories 2 (Recruitment and selection), category 4 (Remuneration and Benefits). In general, there is a lack of autonomy in the HRM area, a fact that impels it to a standardized and predetermined behavior in the face of the problems that occur. Thus, there is a paradox as to the characteristics of HRM in sugar and alcohol companies, because even in search of competitive advantages obtained by better use of people, its policies and practices still seem to be plastered and pre-determined, without opening for questioning and without achieving overcome existing challenges. Therefore, the HRM of the sugar and alcohol sector in this research findings tends to assume a traditional and little strategic role in its subsystems, which are labor relations, recruitment and selection, training and development and remuneration and benefits. Only for the Occupational Health and Safety category and observes a proactive role for HRM, with greater participation in the design of programs and activities focused on the well-being of employees. Therefore, the results indicate that the traditional HRM precepts prevail, culminating in an operational focus as seen in categories one (1), two (2), three (3), and four (4) of the data.”
  • As a contribution of this research, the findings that characterize HRM in agricultural organizations are highlighted now in the Conclusions.

After this extensive effort, we thank you again for your attention. Now, we are aware that we are submitting a much more appropriate piece to be published by Sustainability.

Yours,

The Authors.

Round 2

Reviewer 2 Report

The paper has been improved following the suggestions of the reviewer.